# Molecular Mechanism of Mutational Disruption of DCLK1 Autoinhibition Provides a Rationale for Inhibitor Screening

**DOI:** 10.3390/ijms241814020

**Published:** 2023-09-13

**Authors:** Weizhi Chen, Rui Liu, Yamei Yu, Dongqing Wei, Qiang Chen, Qin Xu

**Affiliations:** 1State Key Laboratory of Microbial Metabolism & Joint International Research Laboratory of Metabolic and Developmental Sciences, School of Life Sciences and Biotechnology, Shanghai Jiao Tong University, Shanghai 200240, China; chenweizhi@sjtu.edu.cn (W.C.); dqwei@sjtu.edu.cn (D.W.); 2Department of Biotherapy, Cancer Center and State Key Laboratory of Biotherapy, West China Hospital, Sichuan University, Chengdu 610041, China; lr1995@stu.scu.edu.cn (R.L.); yamei_yu@scu.edu.cn (Y.Y.)

**Keywords:** DCLK1, cancer mutation, autoinhibitory domain, molecular dynamics simulation, inhibitors screening

## Abstract

Doublecortin-like kinase 1 (DCLK1) is a prominent kinase involved in carcinogenesis, serving as a diagnostic marker for early cancer detection and prevention, as well as a target for cancer therapy. Extensive research efforts have been dedicated to understanding its role in cancer development and designing selective inhibitors. In our previous work, we successfully determined the crystal structure of DCLK1 while it was bound to its autoinhibitory domain (AID) at the active site. By analyzing this structure, we were able to uncover the intricate molecular mechanisms behind specific cancer-causing mutations in DCLK1. Utilizing molecular dynamics simulations, we discovered that these mutations disrupt the smooth assembly of the AID, particularly affecting the R2 helix, into the kinase domain (KD). This disruption leads to the exposure of the D533 residue of the DFG (Asp-Phe-Gly) motif in the KD, either through steric hindrance, the rearrangement of electrostatic interactions, or the disruption of local structures in the AID. With these molecular insights, we conducted a screening process to identify potential small-molecule inhibitors that could bind to DCLK1 through an alternative binding mode. To assess the binding affinity of these inhibitors to the KD of DCLK1, we performed calculations on their binding energy and conducted SPR experiments. We anticipate that our study will contribute novel perspectives to the field of drug screening and optimization, particularly in targeting DCLK1.

## 1. Introduction

Doublecortin-like kinase 1 (DCLK1) is a serine/threonine kinase originally identified in the nervous system [1]. While it was initially believed to regulate the polymerization of microtubules, subsequent research has revealed its involvement in various diseases, such as obesity-induced cardiomyopathy [2] and cancers [3]. However, the precise signaling pathways, substrates, and regulators associated with DCLK1 are not yet well understood.

DCLK1 consists of a tandem doublecortin domain at its N-terminal and a serine/threonine kinase domain (KD) at its C-terminal. The doublecortin domain binds to microtubules and regulates their polymerization [4,5,6]. The kinase domain has a significant similarity to the calmodulin-dependent kinase CaMK. There are four isoforms of DCLK1 associated with different cancers. All isoforms share the same kinase domain but differ in their N-terminal domains and part of their C-terminal regions [7]. In isoforms 2/4, the kinase domain is functionally independent [8] and contains an autoinhibitory domain (AID) that may function as a pseudo-substrate for DCLK1 [9]. The AID consists of 53 residues, including three helix structures: two α-helices (R1 and R2) and one 3_10_-helix (R3) (Figure 1A). R2 is believed to occupy the ATP binding site in the kinase domain, and the peptide of the C-terminal of the AID is thought to occupy the potential peptide substrate-binding site [7].

DCLK1 has been found to play a critical role in the development of cancer. Increased expression and activity of DCLK1 can promote various cancers, such as colorectal cancer [10] and pancreatic cancer [11], and can enhance the clonality, invasiveness, metastasis, and reproducibility of cancer cells [12,13,14]. DCLK1’s effect on cancer is also influenced by its interactions with other proteins. For example, its interactions with the mutated KRAS protein have an important impact on the occurrence of pancreatic cancer [3,15]; it can enhance the pluripotency of intestinal tumors caused by *Apc* gene mutation [16,17]; and a loss of ALDH1B1, which is highly expressed in cancer stem cells of colorectal cancer and pancreatic ductal adenocarcinoma, could lead to the downregulation of DCLK1 gene expression in colon cancer cells [18]. Consequently, DCLK1 has become a target for cancer treatment and a biomarker for early cancer prevention [19]. Various drugs are being developed to inhibit DCLK1 expression or activity in cancer patients, aiming to inhibit tumor growth and cancer cell migration [20].

In our previous study [7], several somatic DCLK1 cancer mutations in the AID and its binding regions in the kinase domain were found from the COSMIC (Catalogue of Somatic Mutations in Cancer) database (Figure 1B–D, Table 1). The upregulation of DCLK1 kinase activity in these mutants were then confirmed by PHOS-Tag electrophoresis and ATP consumption assays. However, the detailed molecular structure and mechanism of autoinhibition on DCLK1’s kinase activity required further investigation.

In this study, we conducted molecular dynamics simulations to simulate the binding of the AID to the kinase domain as a pseudo-substrate. Through these simulations, we revealed a variety of molecular mechanisms of different mutations to affect the binding between the AID and the kinase domain. In addition, we noticed that the salt bridge between K692 near the R3 helix of the AID and D533 of the conserved DFG (Asp-Phe-Gly) motif could be disrupted by some of the mutations, which exposed the aspartic acid and facilitated the ATP binding to KD. Based on these findings, we screened several potential small molecules as DCLK1 inhibitors using database screening, small molecule docking, and binding energy calculations. The interaction modes of these small molecules with DCLK1 may serve as a new approach for drug screening and design targeting DCLK1.

## 2. Results and Discussions

### 2.1. The Autoinhibition by AID Was Simulated Using Molecular Dynamics Simulation in Wild-Type DCLK1

In the crystal structure of DCLK1, the AID and the KD are tightly bound mainly through a stable hydrophobic interaction network (Figure 2A), especially by interactions around the R2 and R3 helixes. In addition, close to the C-terminal of the R3 helix, the K692 residue forms salt bridges with two key residues on the kinase domain, D511 and D533 (Figure 2A). D511 is a catalytic aspartate, which is conserved in all kinases. D533 is of the DFG motif, which is required to coordinate the positions of the magnesium ions and the ATP phosphate groups for catalysis. All these interactions assuring the competitiveness of AID with ATP in its binding site could be basically reproduced in a 200 ns molecular dynamics simulation, during which the AID was assembled back into the KD from the “pre-binding state” (Figure 2B–E, Appendix A). 

### 2.2. G399E and A686T Mutants Hinder the Assembly of the AID

In the crystal structure of DCLK1, the R2 helix of the AID can compete with ATP for the same binding site, thus inhibiting DCLK1 in humans. Therefore, it is reasonable to propose that the mutations occurred near the R2 helix and its binding site would have a more direct and dramatic effect on the kinase activity of DCLK1.

G399 is located on the KD of DCLK1 and adjacent to A686 at the C-terminus of the R2 helix. The G399E mutation has less effect on the R2 helix, and its adjacent hydrophobic interaction regions may hinder the assembly of the AID into the binding site because of its negatively charged long side chain, especially to the peptide from A686 to the C-terminus of the AID, as shown in the simulation results (Figure 3, Appendix A). At the same time, E399 attracted the positively charged K692, curled the AID and dislocated its C-terminus, and exposed the two key residues of D511 and D533, which may facilitate the binding of ATP and reduce the inhibition via AID.

A686 is at the C-terminus of the R2 helix and is spatially adjacent to G399, where the R2 helix and its binding residues form a hydrophobic environment. The A686T mutant in the AID not only induces a steric hindrance by the longer side chain, similarly to G399E, but also brings a hydrophilic group that is unfavorable to R2 binding. In the 200 ns MD simulation on A686T mutant (Figure 4, Appendix A), the AID did not assemble back into to the binding site, which was mainly due to the obstruction of R2 and R3 helices into the KD by the side chain of T686, whereas other parts of AID have little impact from this mutation. At the same time, this mutation disrupts the two salt bridges of K692 with D511 and D533, and exposes these two residues. Although the A686T mutation has little effect on the integrality of AID, the change of hydrophobic properties of residues increases the flexibility of the AID in the simulation process.

The conclusions above were further confirmed by analyses on the binding energies between the AID and the KD calculated from the simulation trajectories (Table 2). Compared with the wild type, the average binding energies of the G399E and A686T mutants are both importantly increased, which is consistent with the disrupted AID binding. However, the change of the binding energies could be mainly attributed to the loss of contributions from the residues at the C-terminal of the AID (A686-R701) in the G399E mutant, and from the residues close to the R2 and R3 helices (V682-R694) in the A686T mutant (Figure 5), suggesting that the binding of different segments of AID is more importantly disrupted.

The simulation results of the G399E and A686T mutants confirmed our earlier assumption that the mutations of residues at the AID binding site into larger ones may hinder the autoinhibition of DCLK1, which could be attributed to a combination of steric hindrance and altered residue properties. As a result, the AID, especially the R2 and R3 helixes, is dissociated outside the KD, which is consistent with the failure to obtain their crystals in the previous experiments. Furthermore, the release of the AID from KD exposes the ATP binding site, which may be one of the reasons for the increase in the kinase activity of DCLK1 and the development of cancer.

### 2.3. G681E and P675L Mutants Disrupt the Structural Stability of the AID

G681 is the N-terminal to the R2 helix of the AID, but does not form a part of the helix. It is worth noting that three typical β-turn structures are found in the K674-G681 peptide, and we named this peptide next to the R2 helix as the “multi-turn region”, in which the three β-turn structures from the N-terminus to the C-terminus are named as turns 1–3, respectively. The canonical proline P675 in turn 1 is also a cancer mutation site found in the COSMIC database. Both the P675L mutant and the G681E mutant could vary the backbone conformation, destroy the local β-turn structure, and unwind the R2 helix, as shown in the simulation results (Figure 6B,C). Consistently, in the only mutant crystal structure obtained from the crystallization experiment, the P675L mutant has most of the electron density missing in the multi-turn area, which suggests the partial unfolding in this region that may disrupt the AID binding [7].

### 2.4. Mutations near the R1 Helix Have Relatively Minor Effects

In our earlier study [7], we proposed that the S660L mutant located in the R1 helix of the AID had less of an increase in kinase activity, considering that it is far from the binding site of the R2 helix and ATP in the catalytic core. Consistently, we found that this mutation in the R1 helix had a relatively weaker effect on the binding of the R2 region via the MD simulations (Figure 7).

### 2.5. The Mutations Reduce the Binding between the AID and the KD of DCLK1 through Diverse Molecular Mechanisms 

DCLK1 has been found to be associated with a variety of cancers, where its increased expression/activity is one of the important reasons. The COSMIC database has several cancer mutations in the AID and the AID-binding regions of DCLK1. In our earlier study [7], the crystal structure of DCLK1 containing the AID at the C-terminal was crystallized. The AID was competitive with ATP by binding to the ATP binding site of the KD as a pseudosubstrate, thus inhibiting the kinase activity of DCLK1. Through the analysis of cancer mutations occurring near AID, it was found that these mutants showed a higher autophosphorylation activity. Therefore, it was proposed that mutations with larger side chains interfered with AID binding, relieved the autoinhibition of DCLK1, and led to the increased DCLK1 kinase activity. However, the crystal structures of most of the DCLK1 mutants could not be obtained, which make it difficult to explain the molecular mechanism through detailed structures. Using molecular dynamics simulations, we were able to reveal more details on the mechanisms of the mutations to relieve the autoinhibition of DCLK1.

In this study, the “pre-binding state” was constructed as the initial structure through a manual adjustment of the crystal structure of wild-type DCLK1, and the molecular dynamics simulations reproduce the automatic assembly of the AID into the KD, leading to the “AID-binding state”. In DCLK1, the AID is complementary to the ATP-binding pocket and binds with the KD mainly through a series of hydrophobic interactions. Therefore, changes of the residue properties, the size of the side chains, and the influence of the flexibility of the AID may all affect the binding ability of the AID to the KD. Particularly, the R2 helix occupies the ATP binding site, so the mutation occurring near it may be more significant. We used the same method to construct and simulate the mutant system, confirmed the previous experimental results of mutants, and explained the molecular mechanism in detail.

As shown in the summary figure (Figure 8), the G399E mutant disrupted the assembly of the AID through the steric resistance and electrostatic attraction, and the A686T mutant made the R2 helix free from the KD by interfering with the hydrophobic interaction between the AID and the KD. The effect of these two groups of mutations is not only from the larger side chains, but also due to changes in the properties of residues, such as hydrophilicity (A686T mutant) and electrostatic property (G399E mutant). On the other hand, the G681E and P675L mutants surprisingly disrupted the local structure of the “multi-turn region”, leading to a too flexible AID to assemble into the ATP binding site in the short time. Lastly, the S660L mutant had a relatively insignificant effect possibly because its mutation site was neighbor to the R1 helix, relatively far from the R2 region.

### 2.6. Screening and Analyzing of Possible DCLK1 Small-Molecule Inhibitors Based on the Binding Molecular Mechanisms Revealed by the Cancer Mutations

Considering the important role of DCLK1 in cancer, great efforts have been placed in drug screening targeting in relation to it, especially from other kinase inhibitors. By far, several small-molecular inhibitors of DCLK1 have been found, as shown in Table 3. Patel Onisha et al. [28] screened NVP-TAE684, an inhibitor of ALK using thermal shift assay (TSA); Jang Dong Man et al. [29] screened Ruxolitinib, an inhibitor of JAK using the method of differential scanning fluorimetry (DSF) assay; and Ferguson Fleur M et al. [30] developed DCLK1-IN-1 using chemoproteomic profiling. Although the inhibitors originally targeting on other kinase may have specificity problems to be directly used as a drug for DCLK1, the results of all these attempts showed their good affinity to DCLK1, which means that they could be good leads for further screening from other chemical databases or templates for further drug design. In the screening of these inhibitors, the interactions around D475 were emphasized for their good affinities. However, Patel Onisha et al. [28] also crystallized DCLK1 with AMPPN, an ATP analog (Appendix A), whose phosphate group forms a hydrogen bond with D533 in the crystal structure, suggesting D533, another possible anchoring residue, for small-molecule inhibitors.

Similar to the AMPPN inhibitory complex (5JZJ), in our crystal structure of DCLK1 containing the AID (6KYQ), the R2 helix occupies the major ATP binding site, and the salt bridge between K692 and D533 provides one of the important interactions for the binding of the AID. Consistently, in the molecular mechanisms revealed by our MD studies, the loss of this salt bridge in the A686T and G399E mutation directly disrupts the binding of R2 to the KD, and thus lowers the inhibitory function of the AID. Interestingly, although no special attention has been paid to the hydrogen bonding between the small molecules and D533 in the earlier screening, it exists in the binding of NVP-TAE684, whose IC50 value is lower than other small molecules (Table 3). Therefore, it is reasonable to assume that the compound that forms hydrogen bonds with D533 may be more competitive to inhibit ATP binding, suggesting this interaction to be a promising condition for the preliminary screening of DCLK1 inhibitors.

We used the binding mode of DCLK1 to AMPPN as a template to perform an IFP search for kinase–ligand complexes in the KLIFS database, where the similarity was limited to 0.75 and above. Through preliminary screening, we obtained 14 different kinase–ligand complexes with their interaction patterns similar to DCLK1-AMPPN (Appendix A). Then, the small-molecule ligands were extracted from the kinase–ligand complexes, and docked into the KD of DCLK1 individually. Only four small-molecule ligands formed hydrogen bonds with D533 of the DGF motif, (Appendix A), whose docking complexes with the KD of DCLK1 were run for 10 ns of molecular dynamics simulations. Then, the binding energies between the ligands and DCLK1 were calculated, which were compared with the binding energies with their original kinase as well as those between DCLK1 and known small-molecule inhibitors, including AMPPN (Table 4). In these four ligands, X9G and 6ZV are two small molecules with relatively lower binding energies. Consistently, structural analyses on the final structure of the simulated docking complexes found that both of them can form stable hydrogen bonds with D533, which may be beneficial to compete with ATP as inhibitors (Figure 9).

In these four ligands, X9G and 6ZV are two small molecules with relatively lower binding energies. As shown in the simulation results (Figure 9), both of them can form stable hydrogen bonds with D533, which may be beneficial to compete with ATP as inhibitors. We used surface plasmon resonance (SPR) to analyze the affinity of 6ZV (Abemaciclib) and NVP-TAE684 with DCLK1, as shown in Figure 10. These two small molecules that form hydrogen bonds with D533 both show good affinity with DCLK1. In addition, the KD value of 6ZV was measured to be 1.92 μM, much lower than that of NVP-TAE684 (19.2 μM), which further confirms the possibility of screening small-molecular inhibitors based on D533 bonding. In summary, this structural feature provides an alternative site for the efficient screening of new small-molecule inhibitors.

At last, we found 6ZV as a commercially available drug, Abemaciclib. Thus, we performed surface plasmon resonance (SPR) experiments to quantify its binding affinity with DCLK1, using the known DCLK1 inhibitor NVP-TAE684 as the positive control. As shown in Figure 10, these two small molecules both show good affinity with DCLK1. In addition, the measured KD value for 6ZV and DCLK1 (1.9 μM) was about 10-fold lower than that for NVP-TAE684 and DCLK1 (19.2μM), respectively, which suggests 6ZV (Abemaciclib) to be a comparable or even better inhibitor for DCLK1. In summary, both the computational prediction and the experimental validation confirmed the structural feature of D533 bonding as an efficient way for the screening of new small-molecule inhibitors for DCLK1.

## 3. Materials and Methods

### 3.1. Construction of the Initial Models

In this study, the crystal structure of isoform 2 of DCLK1 [7] with the PDB ID as 6KYQ was used to construct the initial model of wild-type DCLK1 under the autoinhibited state, where its C-terminal AID is bound with the active site of KD; thus, it is called the “AID binding state”. To simulate the assembly of the AID into the KD of DCLK1, we shifted the AID out of the pocket by about 8.5 Å, and called it “pre-binding state”. These two models were revised into other models of single-site mutants for further MD simulations as below by PyMOL.

### 3.2. Molecular Dynamics Simulations and Binding Free Energy Calculations on the Apoenzyme of Full-Length DCLK1

Amber 2018 was used to run molecular dynamics simulations, using ff03CMAP as the force field for proteins and TIP4P-Ew as the water model for a water box with the side length of 12 Å. The non-bonded cutoff was set to 8 Å. The maximum cycles of energy minimization were set as 6000, in which the first 3000 cycles utilized the steepest descent algorithm before shifting to the conjugate gradient algorithm for the remaining cycles. In the heating process, the initial temperature is 0 K and the reference temperature was set to 298.15 K. Langevin thermostat was selected for temperature control and the collision frequency gamma was set to 2 picoseconds. The time step is set to 0.002 ps. After a 25,000 steps heating process, a 50,000 steps temperature balance process and a 50,000 steps equilibrium process, the molecular dynamics simulation was performed with a running time of 200 ns. The last frame of the simulation process was selected as the representative ending structure for analysis. The simulated trajectory was drawn with an interval of 5 steps, and the RMSD (Root-Mean-Squared Deviations) and binding free energy were calculated based on the drawn trajectories using the MPI (Message Passing Interface) version of MMPBSA.py, where the AID was selected as the “ligand” and the rest of protein (KD) was defined as the “receptor”.

### 3.3. Screening, Docking, and Molecular Dynamics Simulations on the Inhibitor Complexes of the Kinase Domain of DCLK1

KLIFS [31] (Kinase–Ligand Interaction Fingerprints and Structures) is a kinase database that dissects experimental structures of catalytic kinase domains and the way kinase inhibitors interact with them. It contains the structural information of the KD of DCLK1 together with its known small-molecule ligands, including the ATP analog AMPPN [28] and inhibitors NVP-TAE684 [28], Ruxolitinib [29], XMD8-85, and DCLK-IN-1 [30] (corresponding to PDB ID: 5JZJ, 5JZN, 7F3G, 7KX6, respectively). We used the IFP (Interaction Fingerprint) search on KLIFS to screened out other ligand–kinase complexes with similar interaction patterns, where the pattern between DCLK1 and AMPPN was selected as the template and the similarity was limited to 0.75 and above. Then, the small-molecule ligands in these complexes were docked onto the KD of DCLK1 by Autodock, where AMPPN was selected as the template for docking reference. Further screening based on the binding mode of D533 salt bridge were carried out after docking. Thens we carried out the molecular dynamics simulations for these small molecules with DCLK1 and their original protein under the same conditions.

The Antechamber package in AmberTools was used to create the LEaP file for the small molecules, with the “general AMBER force field (GAFF)” for the small-molecule complex system. The protocol for MD simulation is similar to those used in simulations on the apoenzyme of full-length DCLK1, except that the non-bonded cutoff was set to 16 Å and the maximum amount of minimization cycles of energy minimization were set as 200, in which the first 50 cycles utilized the steepest descent algorithm before shifting to the conjugate gradient algorithm for the remaining cycles. After a 1000-steps equilibrium process, the molecular dynamics simulation with a running time of 10 ns was performed. When calculating the binding free energy, the MPI version of MMPBSA.py was selected, and the small molecule was selected as the “ligand”, and the protein (KD) was defined as the “receptor”.

### 3.4. Surface Plasmon Resonance (SPR) Analysis

The expression and purification of DCLK1 kinase domain were same as those described in the Appendix A in our previous work [7]. Briefly, DCLK1 kinase domain was overexpressed in *Escherichia coli* strain Rosetta (DE3) (Novagen, Darmstadt, Germany) and purified sequentially by nickel affinity chromatography and gel filtration chromatography. SPR experiments were performed using a Biacore X-100 system (Cytiva, Washington, DC, USA) at room temperature, with a running buffer of 10 mM Na_2_HPO_4_, 1.75 mM KH_2_PO_4_, pH 7.2~7.6, supplemented with 0.137 M NaCl, 2.65 mM KCl, 0.05% Tween-20, and 5% DMSO. DCLK1 kinase domain (KD) was covalently immobilized onto a CM5 sensor chip using the Amine coupling kit (GE Healthcare). The immobilization of KD resulted in ~12,000 response units (RU). Small-molecule inhibitors were injected over the flow cells at a range of eight concentrations prepared by serial two-fold dilutions, at a flow rate of 30 μL min^−1^, with an association time of 120 s and a dissociation time of 180 s. The data were fitted to a 1:1 binding model so as to calculate equilibrium dissociation constants (binding affinity, *K*_D_) using GraphPad Prism 8.

## 4. Conclusions

In this study, we constructed a series of cancer mutants whose mutations occurred in the AID and its binding site of DCLK1, and provided detailed molecular mechanisms for these cancer mutations through molecular dynamic simulations. The A686T and G399E mutants disrupt the assembly of the AID through the larger side chains and the changes in the properties of residues, while the G681E and P675L mutants surprisingly disrupted the local structure of the “multi-turn region”, leading to a too flexible AID to assemble into the ATP binding site. The S660L mutant has a relatively minor effect. At the same time, according to the importance of the hydrogen bond between D533 and K692, we proposed a new way for screening DCLK1 small-molecule inhibitors, that is, focusing on the interaction between the small molecules and D533. A number of candidate small molecules were found from the KLIFS database, whose combinations with DCLK1 were further tested using binding energy calculations and SPR experimentation. Our research provided a valuable understanding of the cancers caused by DCLK1 mutations and a new way for the development of the drugs targeting it.

## Figures and Tables

**Figure 1 ijms-24-14020-f001:**
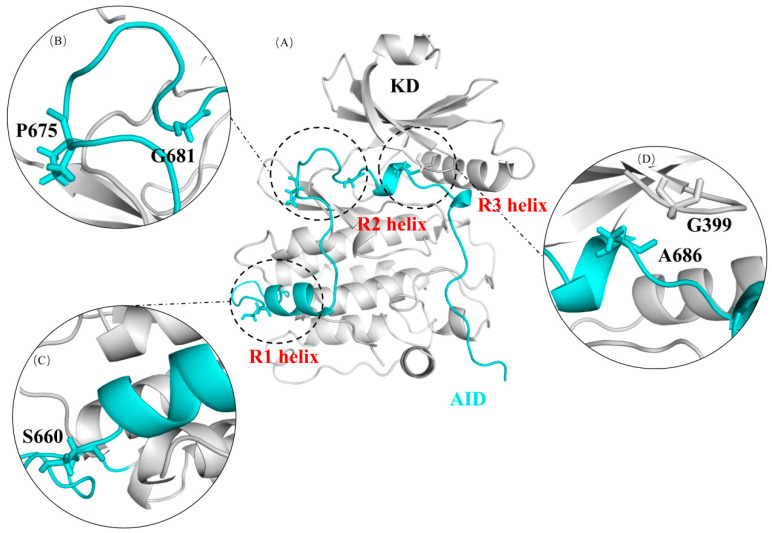
The crystal structure of DCLK1 and the distribution of cancer mutation sites associated with the AID binding. (**A**) The kinase domain (KD) and the autoinhibitory domain (AID) is shown in grey and cyan, respectively, in the structure of the DCLK1; (**B**) G681 and P675 are located near the N-terminus of R2; (**C**) S660 is located near the R1 helix; (**D**) A686 is located on the R2 helix, and its adjacent residue G399 is located in the AID binding site of the KD.

**Figure 2 ijms-24-14020-f002:**
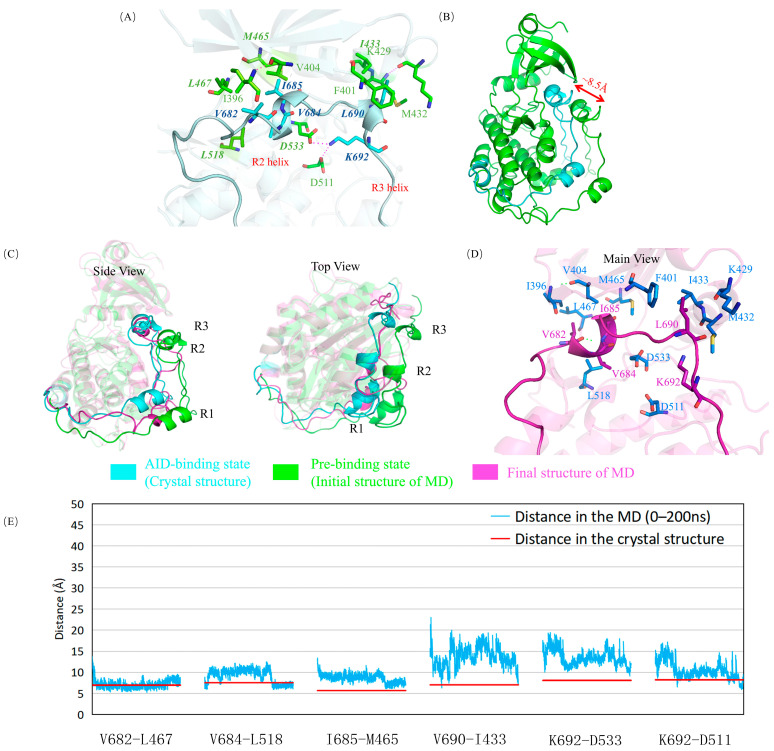
The autoinhibition in wild-type DCLK1. (**A**) Interactions of R2 and R3 helices and their adjacent residues of the KD in a DCLK1 crystal. The four hydrophobic residues (V682, V684, I685, A686T) of the R2 helix on the AID form a stable hydrophobic interaction network with the hydrophobic groups like I396, V404, M465, L467, L518 on the KD, and L690 of the R3 helix forms stable hydrophobic interactions with F401, M432, I433, and other hydrophobic groups of the KD. The K692 residue on the AID forms salt bridges with D511 and D533. The residues of AID and KD are colored in cyan and green, respectively. (**B**) Superposition of the crystal structure of DCLK1 and the initial structure of the molecular dynamics simulation on the AID assembly. The crystal structure (AID-binding state) is shown in cyan, and the initial structure of the molecular dynamics simulation (pre-binding state) is shown in green, where AID was shifted out by about 8.5 Å. (**C**) Superposition of the “AID binding state” (in cyan), “pre-binding state” (in green), and the final structure after simulation (in magenta). The cyan and magenta structure can roughly coincide, especially near the R2 helix, which confirms the assembly of the AID into the KD in wild-type DCLK1. (**D**) The detailed structure of the R2 and R3 helices and their surrounding residues after simulation. The hydrophobic network between AID and KD can be kept relatively stable and consistent to the crystal structure. (**E**) The variation of the distance between Cα atoms of selected amino acids in the simulation process (blue) are compared with those in the crystal structure (red), where the deviations are generally lower than those in the mutant systems (see below), especially at the ending stage of the simulation when the AID has been assembled into the KD.

**Figure 3 ijms-24-14020-f003:**
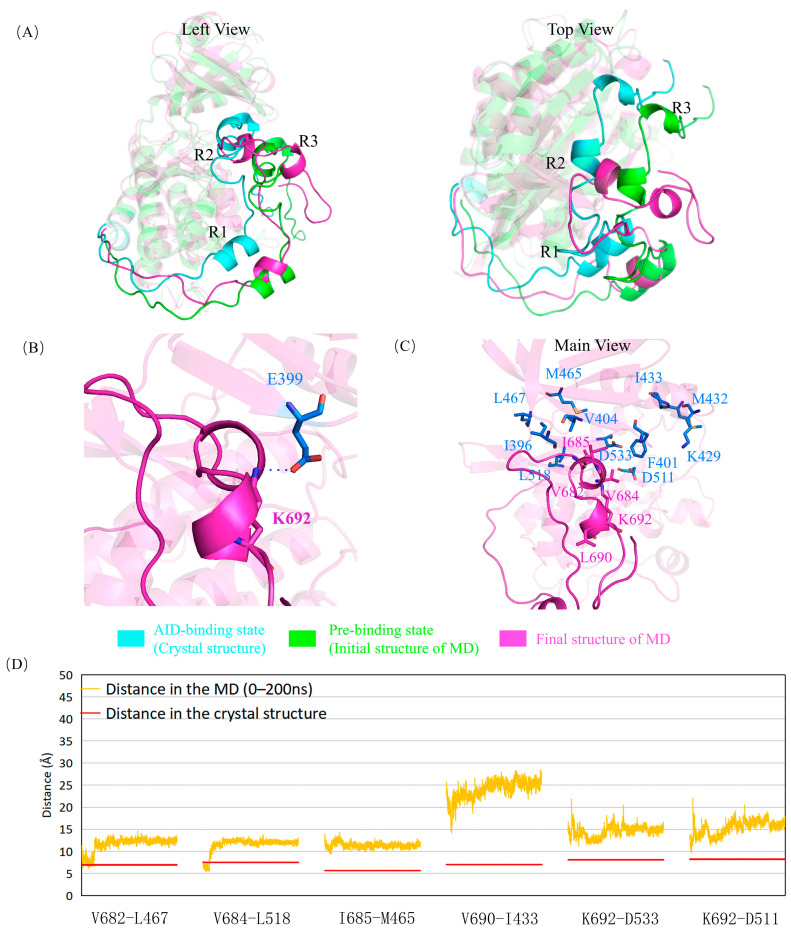
G399E mutant obstructs the assembly of the AID by steric hindrance and electrostatic attraction. (**A**) Superposition of the “AID binding state” (cyan), “pre-binding state” (green), and the final structure after simulation (magenta). Not like the wild-type enzyme, G399E mutant has the AID totally released. (**B**) Electrostatic attraction via E399 with the AID. (**C**) The structure of the R2 and R3 helixes and their surrounding residues after simulation. The original hydrophobic network is destroyed, and some residues, including D533, are exposed because of the mutation. (**D**) Variations of several key-residue distances related to AID–KD interactions. The distances between Cα atoms in the molecular dynamics simulation (orange) are compared with those in the crystal structure (red). The deviations are generally much higher than those in the wild type, especially near the R3 helix (V690-I433).

**Figure 4 ijms-24-14020-f004:**
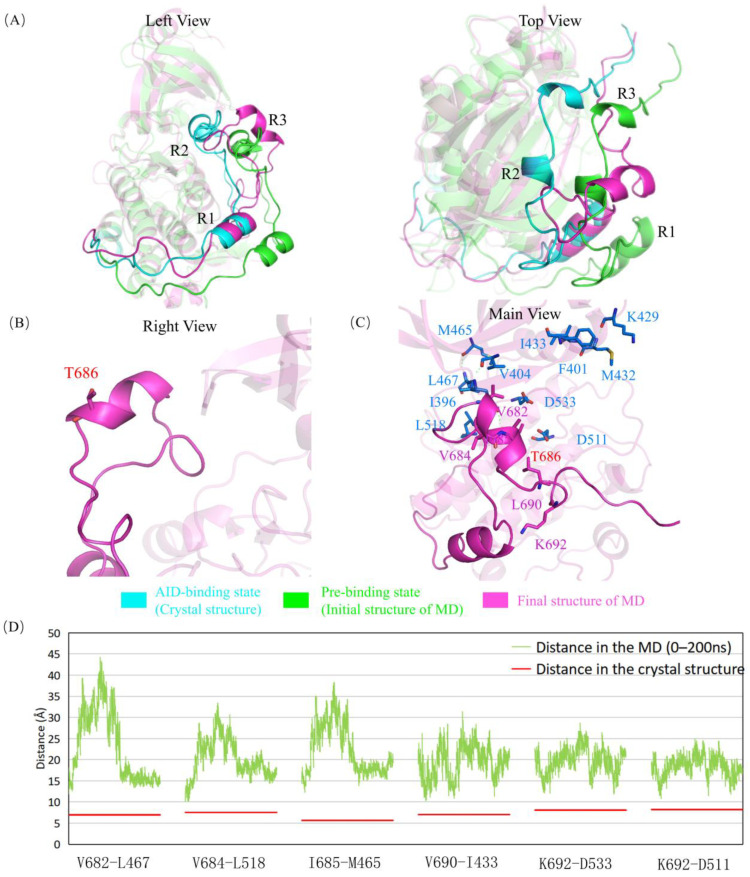
A686T mutant hinders the assembly of the AID through the steric resistance and direct destruction of the hydrophobic network. (**A**) Superposition of the “AID binding state” (cyan), “pre-binding state” (green), and the final structure after simulation (magenta). After the simulation, AID is still out of KD, unlike the crystal structure. (**B**) Structural diagram of T686 for the hydrophilicity of AID. The hydrophilic T686 makes it difficult for AID to enter the hydrophobic pocket, and its side chain points outward, confirming this effect. (**C**) The R2 and R3 helices as well as their surrounding residues after the simulation. The whole AID is completely located outside the hydrophobic pocket in the KD, and the hydrophobic network of the DCLK1 is completely destroyed. (**D**) Variations of several key-residue distances related to AID–KD interactions. The distances between Cα atoms in the molecular dynamics simulation (orange) are compared with those in the crystal structure (red). The deviations are all significantly higher than those in the wild type, especially in the R2 helix (V682-L467, V684-L518, I685-M465).

**Figure 5 ijms-24-14020-f005:**
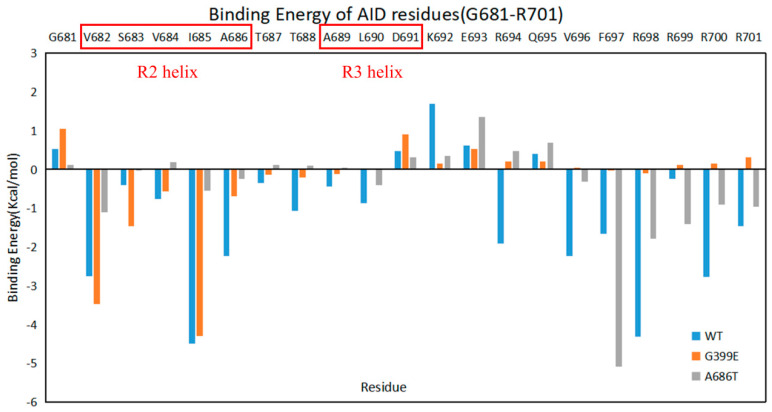
Average contribution to the AID–KD binding energies from the residues on the fragment (G681-R701) of the AID. The residue of wild type, G399E, and A686T are shown in blue, orange, and gray, respectively.

**Figure 6 ijms-24-14020-f006:**
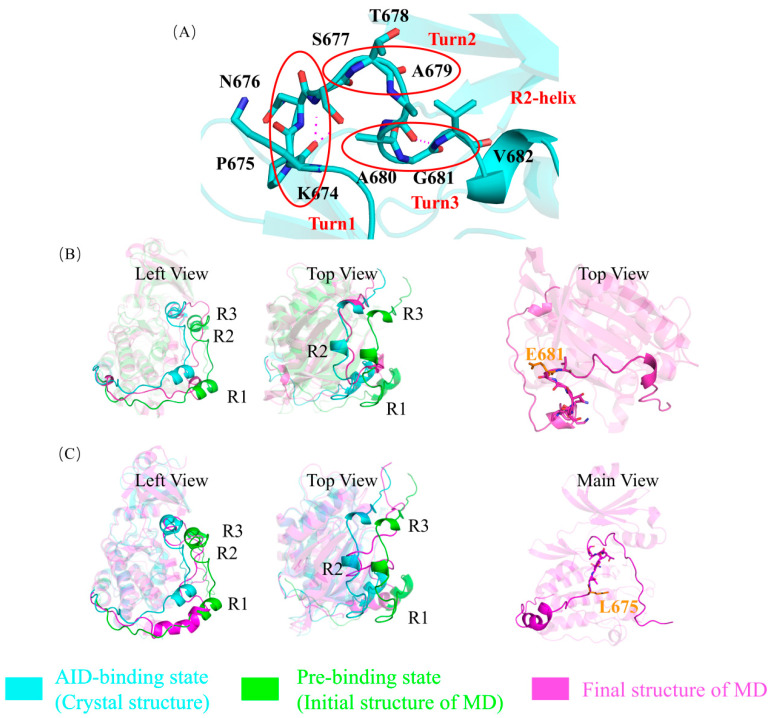
G681E and P675L mutants affect the assembly of the AID by destroying the structural stability of it. (**A**) The “multi-turn region” structure of the AID from the crystal structure of DCLK1 (PDB ID 6KYQ). K674, P675, and N676 form turn 1; S677, T678, and A679 form turn 2; A680, G681, and V682 form turn 3. The “multi-turn region” is located on the N-terminal side of R2 helix. (**B**) Superposition of the “AID binding state” (cyan), “pre-binding state” (green), and the final structure after simulation (magenta) of the G681E mutant. The structural diagram of E681 in the final structure of simulation is shown on the right. (**C**) The superposition of P675L mutant are shown on the left, and the structural diagram of L675 in the final structure of simulation is shown on the right. In the final structure of the two mutants, the most obvious difference with other models is that the R2 helix in the AID has been unscrewed, and the “multi-turn region” has also been destroyed.

**Figure 7 ijms-24-14020-f007:**
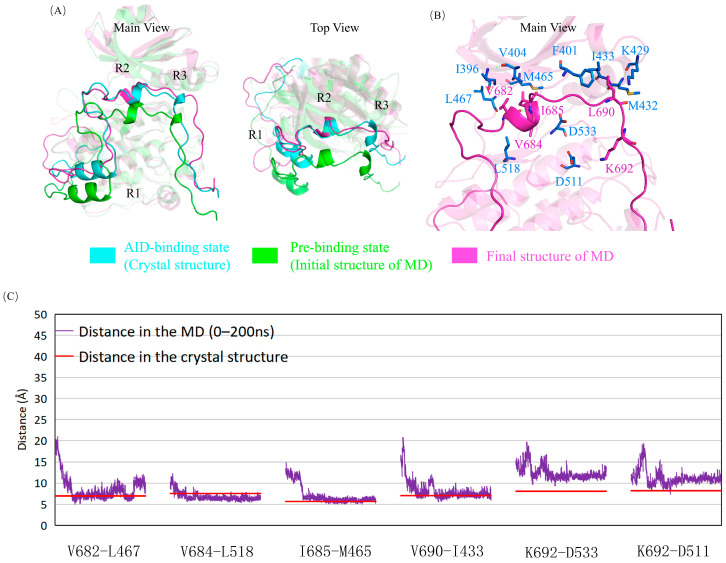
S660L mutation near the R1 helix has little effect on the assembly of the AID. (**A**) Superposition of the “AID binding state” (cyan), “pre-binding state” (green), and the final structure after simulation (magenta). The cyan and magenta structures can roughly coincide, especially near R2 helix, which confirms the little effect of the S660L mutation. (**B**) The structure of the R2 and R3 helixes and their surrounding residues after simulation. The hydrophobic network between AID and KD can be kept relatively stable and consistent with the crystal structure. (**C**) The distances between Cα atoms in the molecular dynamics simulation (purple) are compared with those in the crystal structure (red). In the simulation process, the distances generally drop down quickly to the value comparable with the crystal structure, suggesting that AID is assembled into KD, similar to the wild type.

**Figure 8 ijms-24-14020-f008:**
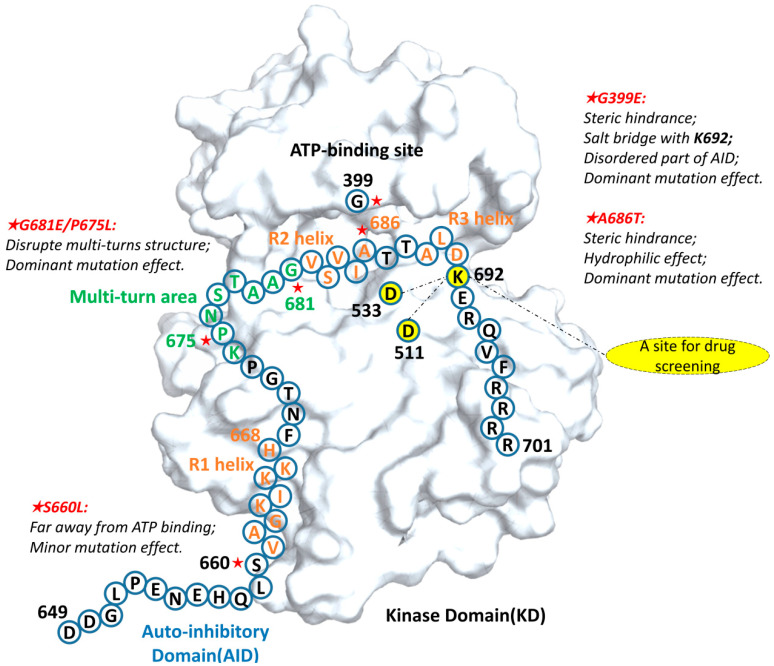
The diverse mechanisms reducing the binding between the AID and the kinase domain of DCLK1. The amino acid sequence and simple structural features of the AID are identified. The green peptides are “multi-turn regions”, the residues marked with “★” are known cancer mutation sites, which occurred in the AID and its binding site in the COSMIC database; the orange peptides are the three helix structures in the AID, and the novel target site for drug screening is highlighted as yellow.

**Figure 9 ijms-24-14020-f009:**
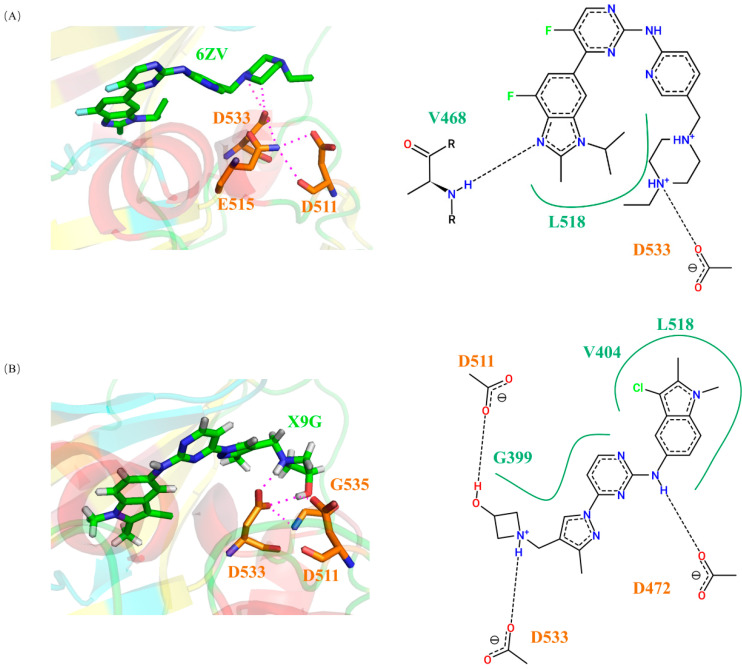
Structures and interaction patterns of DCLK1 binding with X9G and 6ZV. (**A**) At the C-terminal of R2, X9G can form a hydrogen bond network with the residues such as D511, D533, G535, etc. In addition, D472, G399, and L518 can also form hydrogen bonds with X9G; (**B**) 6ZV has relatively fewer interactions with the KD of DCLK1, but it can still form hydrogen bonds with the key residues such as D533, V468, G399, etc., and bind stably with KD.

**Figure 10 ijms-24-14020-f010:**
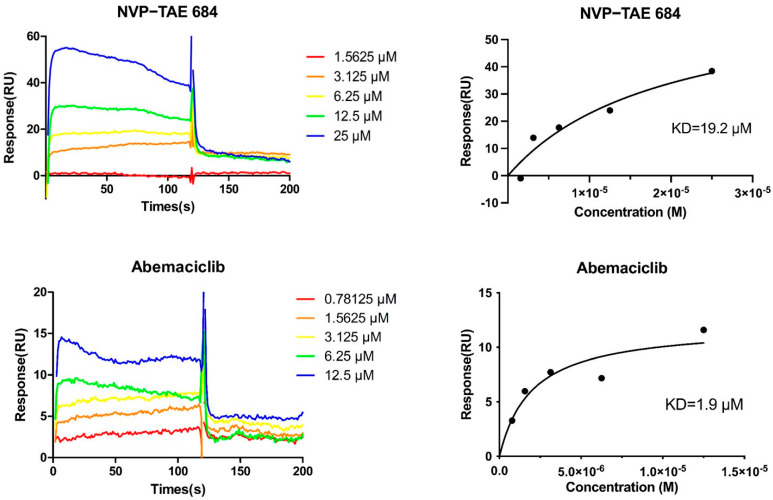
Binding affinity between small-molecule inhibitors and DCLK1 kinase domain measured by SPR. DCLK1 kinase domain is immobilized on the sensor chip and the small-molecule inhibitors are used as the analyte, which is injected at a series of 2-fold serial dilutions. Left, SPR binding curves under defined conditions. Right, the equilibrium response plotted against the analyte concentrations to calculate the KD value.

**Table 1 ijms-24-14020-t001:** Cancer mutations of DCLK1 in the COSMIC database that have been identified clinically and are associated with the autoinhibitory regions.

Mutation Site	Region	COSMIC
S660L	AID-R1	COSP28834 [21]/COSP35781 [22]
I665M	AID-R1	COSP37842 [23]
P675L	AID	COSP37678 [24]/COSP44021 [25]
G681E	AID	COSP28835 [26]
A686T	AID-R2	COSU419
G399E	AID Binding site	COSP40391 [27]

**Table 2 ijms-24-14020-t002:** Average binding energies between the AID and the KD during the MD simulations. The average binding energies between the KD and AID of the G681E and A686T mutants are significantly higher than that of the wild type, which confirmed the disruption on the AID assembly.

Mutant	Average Binding Energy (Kcal/mol)
Wild type	−90.8021
G399E	−38.4060
A686T	−60.9727

**Table 3 ijms-24-14020-t003:** Known small-molecular inhibitors of DCLK1.

Small Molecules	Original Kinase Receptor	Main Experimental Methods	IC50 (nM)	Hydrogen Bond with D533
NVP-TAE684 [28]	ALK	TSA	1084	Yes
DCLK-IN-1 [30]	N/A	chemoproteomic profiling	2346	No
Ruxolitinib [29]	JAK	DSF	1645 (1230–2207)	No

**Table 4 ijms-24-14020-t004:** The binding energies of the kinase–ligand complexes. The binding energies of the four screened small molecules (SGV, 6ZV, X9G, IM9) and the three known inhibitors with DCLK1 (NVP-TAE684, DCLK-IN-1, Ruxolitinib) all have lower binding energies than that of AMPPN. Notably, the binding energy between DCLK1 and NVP-TAE684, the only small-molecule inhibitor hydrogen bonded with D533, is significantly lower than those of other small molecules. Particularly, in the four newly screened potential inhibitors from the KLIFS database, X9G has the binding energy with DCLK1 that is lower than with the original protein, making it a potential lead for the drug development targeting on DCLK1. The unit of binding energy is Kcal/mol and the original kinases of these ligands are shown in the square brackets.

Ligand HET-Code/Name	IFP Similarity (with AMPPN)	Binding Energy with DCLK1	Binding Energy with Original Kinase
SGV	0.76	−35.58	−37.72 [GRK5]
6ZV(Abemaciclib)	0.75	−50.94	−55.37 [CDK6]
X9G	0.82	−51.13	−48.23 [SYK]
IM9	0.79	−42.53	−43.90 [CDK2]
NVP-TAE684		−54.54	
DCLK-IN−1		−49.33	
Ruxolitinib		−41.31	
AMPPN		−34.32	

## Data Availability

The data that support the findings of this study are available from the corresponding author upon reasonable request.

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
