# Peer review of "Molecular Mechanism of Mutational Disruption of DCLK1 Autoinhibition Provides a Rationale for Inhibitor Screening"

_ijms, 2023, doi:10.3390/ijms241814020_

Round 1
Reviewer 1 Report
In the present study, the authors confirmed their previous observation that some DCLK1 mutations may affect the binding of the auto-inhibitory domains and the kinase domain and used this information to screen for novel small molecules that might have an inhibitory role DCLK1 protein. The manuscript’s subject is not innovative, but it is relevant for the scientific community, it is well structured although I would recommend moderate editing of the English language, particularly on the abstract and introduction.
Additionally, I have some suggestion/corrections:
- I would recommend the moderate usage of abbreviatures (e.g. DFG in the abstract; RMSO, MPI, etc). For a non-expert audience, their usage can make the reading of the manuscript a bit bothersome.
- The gene names’ must be italicized. Please correct in the manuscript.
- In the Introduction the following sentence “…Interataction with Kras mutation…”. Do the authors mean “mutated Kras”?
- The following sentence on section 2.4 “The expression and purification of DCLK1 kinase domain were described as previously.” is confusing. “Described as previously” where? There’s a reference missing, or was it described somewhere in the manuscript?
- The identification of small molecules is very relevant, however, the ones that were identified namely Ruxolitinib is a JAK’s inhibitor. I would like to see this aspect acknowledged in the manuscript, in the sense that these inhibitors are not specific for DCLK1 and their usage might trigger other effects.
- During the analysis, did the authors consider the presence of tissue specific DCLK1 isoforms, and the respective impact of the identified mutations? It would be important to acknowledge this aspect.
I would recommend moderate editing of the English language, particularly on the abstract and introduction.
Reviewer 2 Report
In their manuscript Chen et al. constructed a series of cancer mutants whose mutations occurred in auto-inhibitory domain of DCLK1, and provided detailed molecular mechanisms for these mutations through molecular dynamics simulations. The authors screened database and found novel small-molecule inhibitors, and estimated their binding to the kinase domain of DCLK1.
The experimental design has been properly planned and results have been well documented. I have only minor observations:
1) DCLK1 should be more detailed described. What is its physiological function? What type of kinase is it? (serine/threonine kinase or tyrosine kinase). What are the substrates of this enzyme?
2) In the Introduction section the authors wrote: “There has been a number of drugs in development to inhibit the expression or activity of DCLK1 in cancer patients, so as to further inhibiting tumor growth and cancer cell migration [13. Bailey, J.M., Alsina, J., Rasheed, Z.A., et al. (2014). DCLK1 marks a morphologically distinct subpopulation of cells with stem cell properties in preinvasive pancreatic cancer. Gastroenterology 146, 245–256.]”. Which are the drugs? Bailey et al. investigated whether pre-invasive pancreatic neoplasia contains a subpopulation of cells with distinct morphologies and cancer stem cell-like properties. And they concluded that targeting this cell population may have therapeutic potential in the treatment and/or chemoprevention of pancreatic cancer. Is the article no. 13 quoted correct?
3) Moreover the authors wrote that DCLK1 interactions with Kras gene mutation have an important impact on the occurrence of pancreatic cancer. I understand that it is about interactions of the kinase with the protein product of the mutated gene, not directly with the Kras gene. Please correct this sentence in the Introduction section.
Round 2
Reviewer 1 Report
Dear Authors,
All my questions have been addressed.
Thank you.